# Grid Saliency for
# Context Explanations of Semantic Segmentation

**Lukas Hoyer**         **Mauricio Munoz**         **Prateek Katiyar**

**Anna Khoreva**         **Volker Fischer**

Bosch Center for Artificial Intelligence
lukas.hoyer@outlook.com         {firstname.lastname}@bosch.com

## Abstract

Recently, there has been a growing interest in developing saliency methods that provide visual explanations of network predictions. Still, the usability of existing methods is limited to image classification models. To overcome this limitation, we extend the existing approaches to generate *grid saliencies*, which provide spatially coherent visual explanations for (pixel-level) dense prediction networks. As the proposed grid saliency allows to spatially disentangle the object and its context, we specifically explore its potential to produce context explanations for semantic segmentation networks, discovering which context most influences the class predictions inside a target object area. We investigate the effectiveness of grid saliency on a synthetic dataset with an artificially induced bias between objects and their context as well as on the real-world Cityscapes dataset using state-of-the-art segmentation networks. Our results show that grid saliency can be successfully used to provide easily interpretable context explanations and, moreover, can be employed for detecting and localizing contextual biases present in the data.

## 1   Introduction

In many real-world scenarios, the presence of an object, its location and appearance are highly correlated with the contextual information surrounding this object, such as the presence of other nearby objects or more global scene semantics. For example, in the case of an urban street scene, a cyclist is more likely to co-occur on a bicycle and a car to appear on the road below sky and buildings (cf. objects and their context explanations in Fig. 1). These semantic correlations are inherently present in real-world data. A data-driven model, such as a deep neural network, is prone to exploit these statistical biases during training in order to improve its prediction performance. These biases picked up by the network during training can lead to erroneous predictions and impair network generalization (cf. the effect of context on misclassification in Fig. 2). An effective and safe utilization of deep learning models for real-world applications, e.g. autonomous driving, requires a good understanding of these contextual biases inherent in the data and the extent to which a learned model incorporated them into its decision making process.

Saliency methods [4, 5, 6, 7, 8, 9] have become a popular tool to explain predictions of a trained model by highlighting parts of the input that presumably have a high relevance for its predictions. However, to the best of our knowledge the existing saliency methods are mostly focused on image classification networks and thus are not able to spatially differentiate between prediction explanations. In this work, we propose a way to extend existing saliency methods designed for image classification towards (pixel-level) dense prediction tasks, which allows to generate spatially coherent explanations by exploiting spatial information in dense predictions. We call our approach *grid saliency*, which is a perturbation-based saliency method, formulated as an optimization problem of identifying the minimum unperturbed area of the image needed to retain the network predictions inside a target object

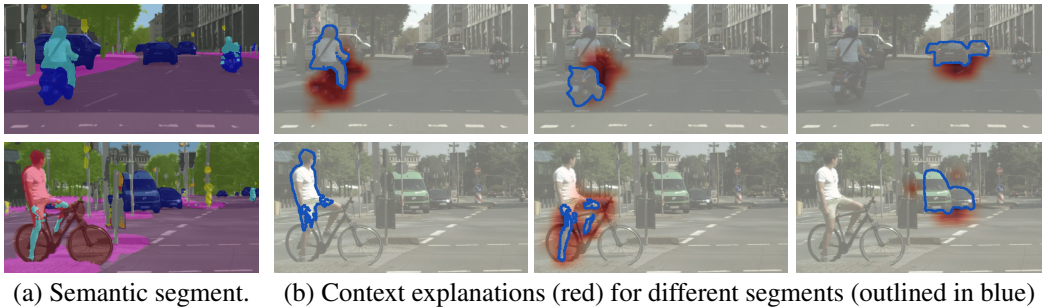

(a) Semantic segment.     (b) Context explanations (red) for different segments (outlined in blue)

Figure 1: Context explanations by grid saliency for semantic segmentation [1, 2] on Cityscapes [3]. Grid saliency not only can contextually explain correct predictions: in the first row the network looks at the motorbike to correctly predict the class rider (light blue); but can also explain erroneous predictions: in the second row the upper body of the rider is incorrectly predicted as person, but for this prediction the bicycle is not salient in contrast to the correctly predicted legs of the rider.

region. As our grid saliency allows to differentiate between objects and their associated context areas in the saliency map, we specifically explore its potential to produce *context explanations* for semantic segmentation networks. The contextual information is known to be one of the essential recognition cues [10, 11, 12], thus we aim to investigate which local and global context is the most relevant for the network class predictions inside a target object area (see Fig. 1 and Fig. 2 for examples).

In real-world scenarios, context biases are inherently present in the data. To evaluate whether the proposed grid saliency is sensitive to context biases and has the ability to detect them, we introduce a synthetic toy dataset for semantic segmentation, generated by combining MNIST digits [13] with different fore- and background textures, for which we artificially induce a context bias between the digit and the background texture. Besides detecting the mere presence of the context bias, we also analyze the ability of our method to localize it in the image (see Sec. 4). We employ this dataset to compare our approach with different baselines, i.e., introduced extensions of gradient-based saliency methods of [4, 14, 15] to produce context explanations. We show that the proposed dataset can serve as a valid benchmark for assessing the quality of saliency methods to detect and localize context biases. We find that gradient-based techniques, in contrast to our grid saliency, are ill-suited for the context bias detection. By design they tend to produce noisy saliency maps which are not faithful to the context bias present in the data, whereas our method has higher sensitivity for context bias and thus can also precisely localize it in the image. We further evaluate grid saliency performance to produce context explanations for semantic segmentation on the real-world Cityscapes dataset [3] and experimentally show that the produced context explanations faithfully reflect spatial and semantic correlations present in the data, which were thus picked up by the segmentation network [1, 2].

To the best of our knowledge, we are the first to extend saliency towards dense-prediction tasks and use it to produce context explanations for semantic segmentation. In summary, our contributions are the following: (1) We propose an extension of saliency methods designed for classification towards dense prediction tasks. (2) We exploit the proposed grid saliency to produce context explanations for semantic segmentation and show its ability to detect and localize context biases. (3) We create a synthetic dataset to benchmark the quality of produced explanations as well as their effectiveness for context bias detection/localization. (4) We investigate the faithfulness of context explanations for semantic segmentation produced by the grid saliency on real-world data.

## 2 Related Work

**Explanations.**   Many methods attempt to interpret the network decision making process by producing explanations via bounding boxes [16, 17] or attributes [18], providing textual justifications [19, 20, 21] or generating low-level visual explanations [4, 22, 5, 6]. Our work builds on top of the latter approaches, also known as *saliency methods*, which try to identify the image pixels that contribute the most to the network prediction. These methods mostly focus on the task of image classification and can be divided into two categories: gradient-based and perturbation-based methods.

Gradient-based methods [24, 4, 25] compute a saliency map that visualizes the sensitivity of each image pixel to a specific class prediction, which is obtained by backpropagating the gradient for the prediction with respect to the image and estimating how moving along the gradient influences the class output. To circumvent noise and visual diffusion in saliency maps, [14] proposes to sum up the

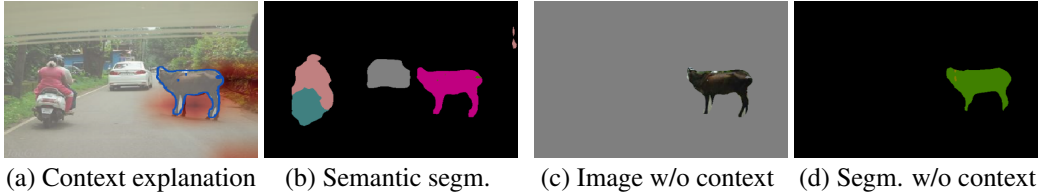

| (a) Context explanation | (b) Semantic segm. | (c) Image w/o context | (d) Segm. w/o context |

Figure 2: Effect of context on semantic segmentation and the context explanation provided by grid saliency for an erroneous prediction, the image is taken from MS COCO [23]. The grid saliency (a) shows the responsible context for misclassifying the cow (green) as horse (purple) in the semantic segmentation (b). It shows the training bias that horses are more likely on road than cows. Removing the context (c) yields a correctly classified cow (d).

gradients over the intensity-scaled input versions, while [15] averages over many noisy samples of the input. Other methods [5, 26, 27] explore integrating network activations into their saliency maps. Gradient-based methods mostly rely on heuristics for backpropagation and as has been shown by [7] may provide explanations which are not faithful to the model or data.

Perturbation-based methods [6, 28, 9, 8] evaluate the class prediction change with respect to a perturbed image, e.g. in which specific regions of the image are either replaced with the mean image values or removed by applying blur or Gaussian noise. The approach in [6] formulates this output change as an optimization problem of the original and perturbed image, while [28] estimates the perturbation mask by training an auxiliary network. To account for image discontinuities, [9, 8] compute the saliency of the masked region by marginalizing it out either over neighboring image regions or by conditioning the trained generative model on the non-masked region and then estimating the classification change. Perturbation approaches might be vulnerable to network artifacts resulting in arbitrary saliency regions [8]. To overcome this, [6, 28] resort to perturbe large image regions.

The above methods are limited to explanations of image classification. In this work, we propose a way to extend them to produce grid saliency maps for dense prediction networks as well (see Sec. 3). We showcase the usability of grid saliency for context explanations of semantic segmentation (see Sec. 4 and 5). The closest related work to produce context explanations is [29], which proposes a network that jointly learns to predict and contextually explain its decisions. In contrast to [29], we focus on the post hoc model explanations and are not limited to the image classification task.

**Semantic segmentation.** CNNs have become a default technique for semantic segmentation [30, 31, 32]. [33] first showcased the use of CNNs for segmentation. Since then, multiple techniques have been proposed, from utilizing dilated convolutions [34, 35] and post-processing smoothing operations [36, 37, 38, 39] to employing spatial pyramid pooling [40, 30, 41, 2] and encoder/decoder architectures [42, 43, 44, 45]. As the accuracy of these methods comes at a high computational cost, there has been an increasing interest in developing real-time semantic segmentation networks with low memory needs [46, 1, 47, 48, 49]. In this work, we aim to produce context explanations for semantic segmentation. To investigate the effectiveness of our approach and show its generalization across architectures, we employ DeepLabv3+ [2] and U-Net [44] with different backbones [1, 50, 51].

## 3 Method

In Sec. 3.1 we introduce the grid saliency method, which allows to produce spatially coherent explanations for dense predictions, and present a way to use it for context explanations of semantic segmentation. Next, in Sec. 3.2 we extend the popular gradient-based saliency methods [4, 14, 15] to produce spatial explanations as well, which we later compare with grid saliency in Sec. 4.

### 3.1 Grid Saliency via Perturbation

Let $f : I \rightarrow O$ denote the prediction function, e.g. a deep neural network, which maps a grid input space $I = \mathbb{R}^{H_I \times W_I \times C_I}$ to a grid output space $O = \mathbb{R}^{H_O \times W_O \times C_O}$, where $W$ and $H$ are the respective width and height of the input and output, and $C_I$ and $C_O$ are the number of input channels (e.g. 3 or 1 for images) and output prediction channels (e.g. number of classes for semantic segmentation). To keep the discussion concrete, we consider only images as input, $x \in I$, and per-pixel dense predictions of the network $f(x) \in O$ as output. The goal is to find the smallest saliency map $M \in [0, 1]^{H_I \times W_I}$ that must retain in the image $x$ in order to preserve the network

prediction in the request mask area $R \in \{0, 1\}^{H_O \times W_O}$ for class (channel) $c \in \{1, ..., C_O\}$. Further on, for simplicity we assume that the input and output spatial dimensions are the same.

Our method builds on top of perturbation saliency methods [6, 28, 9, 8] designed for image classification. They aim to find salient image regions most responsible for a classifier decision by replacing parts of the image with uninformative pixel values, i.e. perturbing the image, and evaluating the corresponding class prediction change. We follow the same image perturbation strategy as [6]. Let $p$ denote a perturbation function that removes information from an image $x$ outside of the saliency $M$. For example, such perturbation function can be the interpolation between $x$ and $a \in I$, where $a$ can be a constant color image, gaussian blur, or random noise. In this case, $p(x, M) = x \circ M + a \circ (1 - M)$. Note, that in practice $M$ operates on a lower resolution to avoid adversarial artifacts [6, 28] and is later upsampled to the original image resolution. In addition, the pixel values of the perturbed image $p(x, M)$ are clipped to preserve the range of the original image space.

With this notation in hand, we can formulate the problem of finding the saliency map $M$ for the prediction of class $c$ as the following optimization problem:

$$M^*(x, c) = \underset{M}{\arg\min} \, \lambda \cdot \|M\|_1 + \| \max(f_c(x) - f_c(p(x, M)), 0)\|_1, \quad (1)$$

where $\| \cdot \|_1$ denotes the $l_1$ norm and $f_c(x)$ is the network prediction for class $c$. The first term can be considered as a mask loss that minimizes the salient image area and perturbs the original image as much as possible. The second term serves as a preservation loss which ensures that the network prediction $f_c(p(x, M))$ for class $c$ on the perturbed image $p(x, M)$ reaches at least the confidence of the network prediction $f_c(x)$ on the original unperturbed image. Thus, the second loss term can be considered as a penalty for not meeting the constraint $f_c(p(x, M)) \geqslant f_c(x)$, hence the use of $\max(\cdot, 0)$ in Eq. 1. The parameter $\lambda$ controls the sparsity of $M$.

We then can spatially disentangle explanations given in the saliency map $M$ for the network predictions in the requested area of interest $R$ from the explanations for the other predictions, by restricting the preservation loss to the request mask $R$ in Eq. (1):

$$M^*_{\text{grid}}(x, R, c) = \underset{M}{\arg\min} \, \lambda \cdot \|M\|_1 + \frac{\|R \circ \max(f_c(x) - f_c(p(x, M)), 0)\|_1}{\|R\|_1}. \quad (2)$$

Further on, we will refer to $M^*_{\text{grid}}$ in Eq. (2) as a grid saliency map.

**Context explanations for semantic segmentation.** We now adapt the grid saliency formulation from Eq. (2) to specifically provide *context explanations* for the requested area of interest $R$. Context explanations are of particular interest for semantic segmentation, as context often serves as one of the main cues for semantic segmentation networks (see Fig. 2). Thus, here we focus on context explanations for semantic labelling predictions and assume that $R$ is the area covering the object of interest in the image $x$. To optimize for salient parts of the object context, we integrate the object request mask $R$ in the perturbation function. For the request mask $R$, the perturbed image $p(x, R) \in I$ contains only the object information inside $R$ and all the context information outside $R$ is removed (with a constant color image $a$). For optimization, we will now use this new perturbed image $p(x, R)$ instead of the maximally perturbed image $p(x, M = 0) = a$ and denote the context perturbation function as $p_{\text{context}}(x, R, M) = x \circ M + p(x, R) \circ (1 - M)$.

The context saliency map for class $c$ and request object $R$ can be obtained via optimization of

$$M^*_{\text{context}}(x, R, c) = \underset{M}{\arg\min} \, \lambda \cdot \|M\|_1 + \frac{\|R \circ \max(f_c(x) - f_c(p_{\text{context}}(x, R, M)), 0)\|_1}{\|R\|_1}, \quad (3)$$

where the saliency map is optimized to select the minimal context necessary to at least yield the original prediction for class $c$ inside the request mask $R$. Note that, $M^*_{\text{context}}$ can be an empty mask if no context information is needed to recover the original prediction inside $R$.

## 3.2 Gradient-Based Variants

Another way to produce spatially coherent saliency maps is to make use of the popular gradient-based saliency methods. Thus, we additionally consider the Vanilla Gradient (VG) [24], Integrated Gradient (IG) [14], and SmoothGrad (SG) [15] saliency methods.

Let $G(x, c) = \partial g_c(x)/\partial x \in \mathbb{R}^{H_I \times W_I \times C_I}$ denote the gradient of the classification network prediction $g_c(x) \in \mathbb{R}$ for class $c$ with respect to the input image $x \in I$. For the classification task, the saliency

maps of VG, IG and SG are computed as:

$$M^{\text{VG}}(x,c) = \sum_{c \in C_I} |G(x,c)|, \quad M^{\text{SG}}(x,c) = \sum_{c \in C_I} \left| \frac{1}{n} \sum_{k=1}^{n} G\left(x + \mathcal{N}(0,\sigma^2), c\right) \right|,$$

$$M^{\text{IG}}(x,c) = \sum_{c \in C_I} \left| \frac{1}{n} \sum_{k=1}^{n} G\left(\frac{k}{n}x, c\right) \right|, \tag{4}$$

where $n$ is the number of approximation steps for IG or the number of samples for SG, and $\mathcal{N}(0,\sigma^2)$ represents Gaussian noise with standard deviation $\sigma$.

Following Sec. 3.1, we next extend the above approaches to produce the saliency $M$ for dense predictions $f_c(x) \in O$ and to spatially disentangle explanations given in the saliency $M$ for the network predictions in the request area $R$ from other predictions. For a given input $x$ and a binary request mask $R$, we denote the normalized network prediction score for class $c$ in the request area $R$ as $S(x,R,c) = \|R \circ f_c(x)\|_1 / \|R\|_1$, $S(x,R,c) \in \mathbb{R}$. Similarly to $G(x,c)$, we define $G_{\text{grid}}(x,R,c) := \partial S(x,R,c)/\partial x \in \mathbb{R}^{H_I \times W_I \times C_I}$ which directly yields $M_{\text{grid}}^{\text{VG/SG/IG}}(x,R,c)$ by replacing $G(x,c)$ in Eq. 4 with $G_{\text{grid}}(x,R,c)$. For the gradient-based context saliency, as in Sec. 3.1 only salient pixels outside of the object area are considered, i.e.

$$M_{\text{context}}^{\text{VG/IG/SG}}(x,R,c) := (1-R) \circ M_{\text{grid}}^{\text{VG/IG/SG}}(x,R,c). \tag{5}$$

Gradient-based saliency maps are prone to be noisy. Thus, to circumvent this and also make them more comparable to the lower resolution perturbation-based grid saliency, in our experiments we apply the spatial mean filter on top of the saliency map with a $(W_I/W_S) \times (H_I/H_S)$ kernel and stride, where $W_S \times H_S$ is the resolution of the perturbation-based saliency map.

## 4 Context Bias Detection on Synthetic Data

Although context biases are inherently present in real-world data, in practice it is hard to measure and alter these correlations and thus to employ this data for benchmarking context bias detection. In order to evaluate whether the grid saliency is sensitive to context biases and has the ability to detect them, in Sec. 4.1 we introduce a synthetic dataset for semantic segmentation with artificially induced context biases.[1] We use this dataset to compare the grid saliency methods proposed in Sec. 3 and show that this dataset can serve as a valid benchmark for assessing the quality of saliency methods for context bias detection and localization, see Sec. 4.2.

### 4.1 Benchmark for Context Bias Detection

**Dataset.** The proposed synthetic toy dataset consists of gray scale images of size $64 \times 64$ pixels, generated by combining upscaled digits from MNIST [13] with foreground and background textures from [52, 53], as can be seen in Fig. 3 (a). In order to introduce different context information for the digit, two background textures are used for the upper and lower half of the image. For each synthetic image a corresponding segmentation ground truth is generated, where the MNIST digit defines the mask and the semantic class (including a background class).

To evaluate the ability of saliency methods to explain context biases, we propose to generate biased and unbiased versions of the dataset. For the unbiased version (DS$^{\text{no-bias}}$), all fore- and background textures appear with equal probability for all digits. For the biased version, a single digit class is coupled with a specific background texture, located randomly either in the upper or lower half of the background. We consider two variants of it, with a weakly DS$^{\text{w-bias}}$ and strongly induced bias DS$^{\text{s-bias}}$. For the dataset with a strongly induced bias DS$^{\text{s-bias}}$, a specific texture appears if and only if a certain digit class is present. For a weakly induced bias DS$^{\text{w-bias}}$, a specific texture always appears along with the biased digit but also uniformly appears with other digits. From the pool of 25 textures, 5 textures are chosen randomly to induce context bias for one of 10 digits. For all 50 texture/digit combinations, a weakly and strongly biased dataset variant with train/test splits is generated.

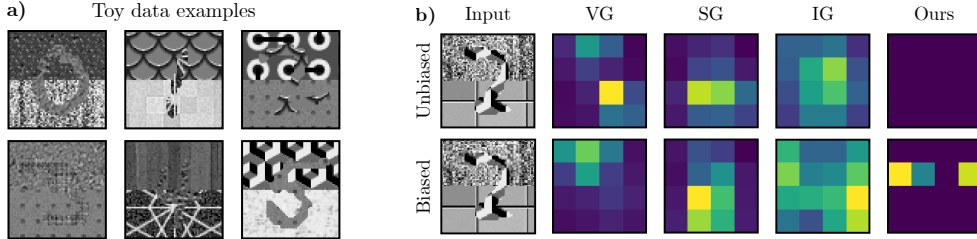

Figure 3: **a)** Synthetic dataset images, see Sec. 4.1 for details. **b)** Context saliency maps of different methods. For the image on the left, the different context saliency maps $M_{\text{context}}$ are shown for the networks [1] trained on the unbiased (top row) and biased (bottom row) dataset versions. For the biased version, the digit *2* is biased with the background texture in the top half of the image. Our perturbation-based grid saliency is able to precisely detect this bias, in contrast to other methods.

**Evaluation metrics.** To evaluate the extent to which a network is able to pick up a context bias present in the training data, we propose to measure on the unbiased $DS^{\text{no-bias}}$ test set the performance of the network trained on the biased $DS^{\text{w/s-bias}}$ training set, using the standard Intersection over Union (IoU, see [33]) metric for semantic segmentation. If the network has picked up the bias, we expect to see a significant drop in IoU for the biased digit segmentation.

To benchmark how well different saliency methods (perturbation- or gradient-based) can detect the context bias of semantic segmentation networks, we propose to evaluate to which extent a context saliency map $M_{\text{context}}(x, R, c)$ (cf. Eq. 3 and 5 in Sec. 3) for the request object mask $R$ and its corresponding class $c$ is concentrated on the ground truth context area $C = 1 - R_{GT}$, where $R_{GT}$ is the ground truth mask of the object $R$, by using a context bias detection metric (CBD):

$$\text{CBD}(x, R, c) = \|C \circ M_{\text{context}}(x, R, c)\|_1 / \|C\|_1. \qquad (6)$$

To benchmark the ability of different saliency methods to localize a context bias, we propose to measure how much of the context saliency $M_{\text{context}}(x, R, c)$ falls into the ground truth biased context area $C_{bias}$, the upper or lower half of the image $x$ by the design of $DS^{\text{w/s-bias}}$, where $C_{\text{bias}}$ is a binary mask of the biased context area. We refer to this metric as a context bias localization (CBL):

$$\text{CBL}(x, R, c) = \|C_{\text{bias}} \circ M_{\text{context}}(x, R, c)\|_1 / \|C \circ M_{\text{context}}(x, R, c)\|_1. \qquad (7)$$

In our experiments we report mIoU and the mean CBD and CBL measures (mCBD, mCBL) per biased digit class $c$, averaging the results across all images in the $DS^{\text{w/s-bias}}$ test set, 5 randomly selected bias textures and 5 different initial random seeds for the training set generation.

## 4.2 Experimental Results

**Implementation details.** We use the U-Net [44] architecture with a VGG16 [51] backbone. As request mask, the segmentation prediction of the target digit is used. The saliency maps with a size of $4 \times 4$ are optimized using SGD with momentum of 0.5 and a learning rate of 0.2 for 100 steps starting with a 0.5 initialized mask. A weighting factor of $\lambda = 0.05$ is used (see Eq. 3). A constant color image is used for perturbation. Further implementation details are provided in the supp. material.

**Bias in the trained network.** We first investigate if the networks trained on $DS^{\text{w-bias}}$ and $DS^{\text{s-bias}}$ have picked up the induced weak and strong context biases in the training data. For this purpose, we evaluate their performance in terms of mIoU on the unbiased dataset $DS^{\text{no-bias}}$ and report the results in Fig. 4 (a), which visualizes the digit-wise mIoU with respect to the biased digit. The first row of the heat map (labeled as N) in Fig. 4 (a) shows the performance of the networks trained on $DS^{\text{no-bias}}$. We observe a clear drop in performance for biased digits (diagonal elements) in comparison to the first row. As expected, the performance drop is higher for the stronger bias. Moreover, the mIoU of the unbiased digits (non-diagonal elements) is also affected by the introduced bias. For example, inducing a bias for the digit nine leads to a decreased performance for the digit four (see second row in Fig. 4 (a)). We observe that this effect mostly occurs for the similar looking digits and, most likely, is caused by the fact that on the unbiased dataset the bias textures also occur with the unbiased digits, resulting in the confusion of similar looking digits for the network. From the observed mIoU drop for biased digits we can conclude that the networks have picked up the introduced bias. However, in real world it is often impossible to collect fully unbiased data. Thus, we next evaluate the ability of our grid saliency to detect context bias only using the biased data.

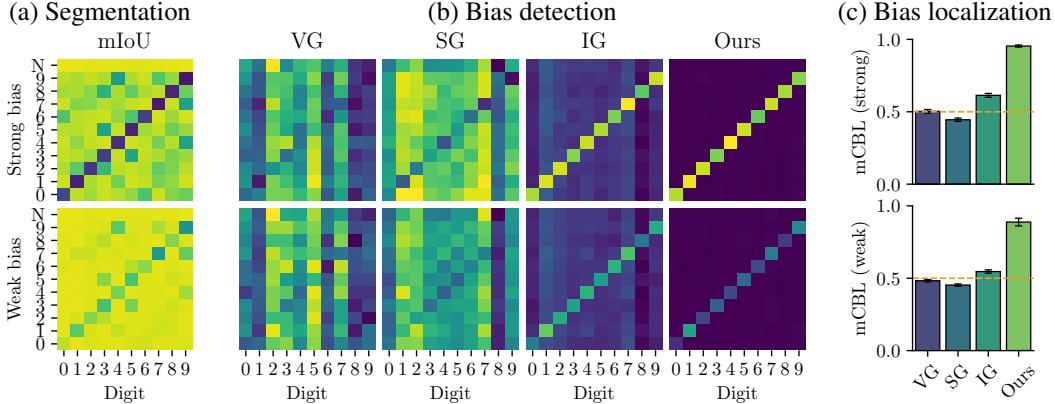

Figure 4: Context bias detection/localization of different saliency methods. **(a)** shows the segmentation mIoU and **(b)** & **(c)** report the context bias detection and localization results, for the strongly and weakly biased datasets, respectively. In **(a)**-**(b)**, the $y$ axis indicates the network bias towards the digit (0-9), with $N$ denoting the unbiased setting. The $x$ axis indicates the corresponding digit result. In contrast to VG [24], IG [14], and SG [15], our grid saliency is able to accurately detect (see diagonal elements in **(b)**) and localize **(c)** the induced context bias, see Sec. 4.2 and suppl. material for details.

**Context bias detection with grid saliency.** In the last column of Fig. 4 (b), we report the context bias detection results for our perturbation-based grid saliency method, described in Sec. 3.1, using the CBD metric (see Eq. 6). The mCBD values are visualized with respect to networks trained on data biased to different digits $DS^{s/w\text{-bias}}$ (y-axis) and for the different digit classes (x-axis) in the biased test set. The only exception is the first row (labeled as N), where for comparison we show the results with no bias, for the network trained on $DS^{no\text{-bias}}$. We observe that our grid saliency shows substantial evidence of context bias for digits with induced bias (diagonal elements), both strong and weak. Even for the weak bias in Fig. 4 (b) the grid saliency still clearly differentiates between biased and unbiased digits (diagonal vs. non-diagonal elements). Note that the bias detection using the grid saliency does not require an unbiased test set. In the suppl. material, we also study the influence of the bias texture as well as the choice of hyperparameters.

**Comparison across different saliency methods.** In Fig. 4 (b) and (c) we compare our perturbation-based grid saliency with the context saliency extensions of gradient-based methods, i.e. VG, SG, and IG (see Eq. 4), using mCBD and mCBL metrics proposed in Sec. 4.1. From Fig. 4 (b) we see that VG and SG are not able to reliably detect the context bias, while IG achieves a comparable performance to our grid saliency. However, in contrast to the perturbation saliency, IG has also high mCBD values for unbiased digits, complicating its use in practice, as one would need to tune a reliable detection threshold, which is particularly challenging for weakly biased data.

**Context bias localization.** In addition to the detection performance, we evaluate how well the saliency methods localize the context bias using the mCBL measure. Fig. 4 (c) shows that VG, IG, and SG have low localization performance, comparable to random guessing ($\sim 0.5$ mCBL), while our grid saliency is the only method which is able to accurately localize the context bias (mCBL above $0.9$) on both strongly and weakly biased data. We evaluate the ability of our perturbation grid saliency to detect and localize context bias across different semantic segmentation networks. For U-Net [44] with the VGG16 [51], ResNet18 [50] and MobileNetv2 [1] backbones, we observe similar bias detection and localization performance (see also supp. material).

## 5   Cityscapes Experiments

Motivated by the success of our perturbation-based grid saliency at detecting context biases on the synthetic dataset described in Sec. 4, we next apply it to Cityscapes [3] in order to produce and analyze context explanations for semantic segmentation in real-world scenes.

**Experimental setup.** We use 500 finely annotated images of the Cityscapes validation set, considering only a subset of classes for the analysis. For our experiments we use the Deeplabv3+ [2] network with a Mobilenetv2 backbone [1]. Our optimization setup largely carries over from Sec. 4.2, with the exception that we optimize a coarse 16 by 32 pixel mask using SGD with a learning rate of 1 for 80 steps and use $\lambda = 0.01$. Additional implementation details are provided in the supp. material.

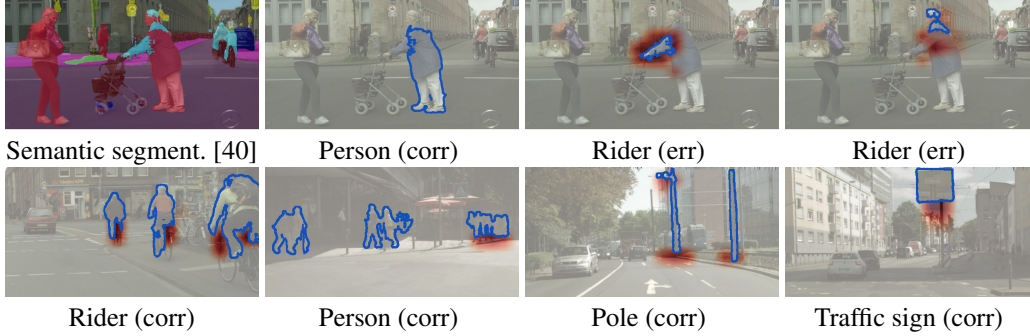

| Semantic segment. [40] | Person (corr) | Rider (err) | Rider (err) |

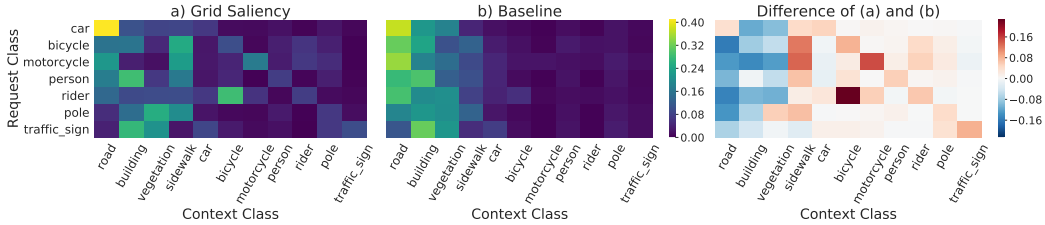

| Rider (corr) | Person (corr) | Pole (corr) | Traffic sign (corr) |

Figure 5: Context explanations by grid saliency for erroneous (err) and correct (corr) semantic segmentation predictions on Cityscapes [3]. Salient red regions visualize the most important context $M_{\text{context}}$ for the requested segment prediction $R$ outlined in blue. See Sec. 5 for discussion.

Figure 6: Class statistics of context explanations on Cityscapes, see Sec. 5 for discussion.

**Analysis of context explanations.** In order to gain a global understanding of the model context bias, we aggregate statistics from the produced grid saliency maps for several requested classes across all validation images. These statistics are summarized in Fig. 6. For each image and class, the context saliency is computed for all sufficiently large instances (at least 10k pixels), as can be seen in the second row of Fig. 5. Next, for each image and semantic class we compute the weighted label distribution of context salient pixels. The weight of each pixel is given by the computed saliency intensity at said pixel and its context class label is taken from the ground truth segmentation. For each requested object class, we group by label across all images and sum the saliency values within each group. This yields an accumulated saliency value for each label. We normalize the results by the sum of all saliency values across all classes and images to yield a probability distribution across labels.

To show that context explanations produced by our grid saliency are meaningful, we additionally compare its accumulated context class statistics with their baseline class distributions. These baselines are computed in the same manner as above, but instead of relying on the optimized saliency map we use a fixed dilation of the contour of all predicted object segments considered in the image, thereby capturing the immediate context around each object in a uniform set of directions.

Fig. 6 compares these accumulated statistics of context explanations for the requested object classes, given on the $y$ axis. We observe that the context explanations are focused across reasonable and somewhat expected class subsets, which vary per class. For instance, in comparison to the baseline, context saliency for the rider class shifts attention from its spatially co-occurring classes such as road, vegetation and building, to mainly the bicycle class. Car context saliency attention is decreased on building and vegetation and mainly focused on road, sidewalk and other cars. Bicycle context saliency mostly attends to sidewalk rather than the road class. Overall, our grid saliency is able to provide sensible and coherent explanations for network decision making, which reflect semantic dependencies present in street scenes. These quantitative findings are validated by the qualitative results in Fig. 5. The second row of Fig. 5 gives several representative examples of context saliencies. Please refer to the supp. material for additional examples and a high-resolution version of Fig. 6 as well as a qualitative model comparison on MS COCO.

**Context explanations of erroneous predictions.** One of the motivations for this work is also to explain unexpected model behavior. Specifically, in case of erroneous predictions (i.e., Fig. 1 and 2), we wish to understand to which extent context bias contributes to the failure. Table 1 shows class statistics of context explanations for correct and erroneous predictions, where the intersection of the ground truth mask and the correct or misclassified prediction region are used as the request mask $R$, and similarly to Fig. 6 semantic class statistics are computed inside the salient context for all sufficiently large request masks (at least 625 pixels). We observe that the saliency severely changes

Table 1: Class statistics of context explanations of correct and erroneous predictions on Cityscapes.

| GT | Prediction | Context class | | | | | | | | | |
|---|---|---|---|---|---|---|---|---|---|---|---|
| | | road | bicycle | motorcycle | car | vegetation | building | sidewalk | rider | person | pole |
| rider | rider | 0.17 | **0.22** | 0.03 | 0.07 | 0.09 | 0.11 | 0.10 | 0.07 | 0.02 | 0.02 |
| | person | 0.13 | 0.10 | 0.01 | 0.05 | 0.10 | **0.24** | 0.05 | 0.08 | 0.11 | 0.04 |
| person | person | 0.12 | 0.03 | 0.00 | 0.09 | 0.15 | **0.22** | 0.13 | 0.01 | 0.05 | 0.05 |
| | rider | 0.09 | **0.18** | 0.02 | 0.04 | 0.10 | 0.13 | 0.07 | 0.08 | 0.15 | 0.03 |
| | car | 0.19 | 0.01 | 0.00 | **0.32** | 0.05 | 0.11 | 0.03 | 0.01 | 0.19 | 0.01 |
| bicycle | bicycle | 0.18 | 0.06 | 0.01 | 0.05 | 0.09 | 0.15 | **0.19** | 0.05 | 0.04 | 0.04 |
| | motorcycle | 0.19 | **0.27** | 0.13 | 0.01 | 0.02 | 0.11 | 0.13 | 0.03 | 0.01 | 0.04 |
| car | car | **0.39** | 0.02 | 0.00 | 0.06 | 0.11 | 0.10 | 0.08 | 0.01 | 0.04 | 0.04 |
| | motorcycle | 0.07 | 0.06 | **0.22** | 0.17 | 0.14 | 0.07 | 0.02 | 0.01 | 0.00 | 0.04 |
| | person | 0.14 | 0.01 | 0.00 | **0.28** | 0.03 | 0.14 | 0.07 | 0.01 | 0.22 | 0.02 |
| motorcycle | motorcycle | **0.18** | 0.04 | 0.10 | 0.04 | 0.07 | 0.14 | **0.18** | 0.05 | 0.03 | 0.06 |
| | bicycle | 0.13 | 0.10 | 0.07 | 0.10 | 0.10 | **0.14** | 0.11 | 0.01 | 0.05 | 0.03 |
| | car | 0.16 | 0.01 | **0.18** | 0.10 | 0.10 | 0.10 | 0.11 | 0.05 | 0.04 | 0.06 |
| pole | pole | 0.04 | 0.02 | 0.00 | 0.05 | **0.28** | 0.23 | 0.10 | 0.00 | 0.03 | 0.04 |
| | building | 0.04 | 0.01 | 0.00 | 0.04 | 0.07 | **0.57** | 0.02 | 0.00 | 0.01 | 0.05 |
| | vegetation | 0.03 | 0.00 | 0.00 | 0.03 | **0.63** | 0.08 | 0.03 | 0.00 | 0.00 | 0.07 |

for the error cases in comparison to the correct predictions. E.g. for the correctly classified rider class, context saliency mostly focuses on bicycle (22%), but is much less present (10%) when rider is mistaken as person and in this case the context saliency shifts from bicycle to building (24%). The opposite effect is observed when person is misclassified as rider. Sidewalk saliency is decreased when bicycle is misclassified as motorcycle (19% vs. 13%) and when pole is misclassified as building (10% vs. 2%). In general, grid saliency is able to provide reasonable explanations of erroneous predictions.

Fig. 1 (second row) and Fig. 5 (first row) showcase how applying grid saliency on the same input image but different prediction outputs is useful for isolating failures caused by context bias. Both of them show examples of context explanations for erroneous segmentations of a single object. In Fig. 5, the arms, head and shoulders of a pedestrian are classified as rider, while the rest of the body is correctly identified as person. The context saliency for the former body parts activates highly on the arms and stroller handles, whereas activations for person does not highlight this support. Thus, a reasonable conclusion is that the misclassification may be attributed to the arm pose and potentially also to the context bias given by the similar appearance of the stroller and bicycle handles.

## 6 Conclusions

We proposed spatial grid saliency, a general framework to produce spatially coherent explanations for (pixel-level) dense prediction networks, which to the best of our knowledge is the first method to extend saliency techniques beyond classification models. We investigated the ability of grid saliency to provide context explanations for semantic segmentation, showing its effectiveness to detect and localize context bias on the synthetic toy dataset specifically designed with an artificially induced bias to benchmark this task. Our results on the real-world data indicated that grid saliency can be successfully employed to produce easily interpretable and faithful context explanations for semantic segmentation, helping to discover spatial and semantic correlations in the data picked up by the network. We hope the proposed grid saliency and the insights of this work can contribute to a better understanding of semantic segmentation networks or other models for dense prediction, elucidating some aspects of the problem that have not been well explored so far.

Besides enabling visual explanations for dense-prediction tasks, we see potential utility of grid saliency for the following applications: *1) Architecture comparison:* Context explanations produced by grid saliency can be used to compare architectures with respect to their capacity to either learn or to be invariant towards context. *2) Network generalization via active learning:* Existing context biases might impair network generalization. E.g., cows might mostly appear on grass during training. A network that was trained and evaluated on this data and picked up that bias will perform poorly in real-world cases, where the cow, for example, appears on road and gets misclassified as the horse class, see Fig. 2. Actions can be taken, such as targeted extra data collection, to improve network generalization. *3) Adversarial detection:* Grid saliency can be used to detect and localize adversarial patches outside object boundaries (e.g. [54]). Cases for which the salient regions lie largely outside an object would strongly indicate the presence of an adversary or misguided prediction. We consider the above-mentioned utilities of grid saliency interesting and promising future research directions.

## Footnotes

[1]The code for generating the proposed synthetic dataset with induced context biases can be found here: `https://github.com/boschresearch/GridSaliency-ToyDatasetGen`.

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
