[Supplementary Material]

# Grid Saliency for
# Context Explanations of Semantic Segmentation

# Supplementary Material

**Lukas Hoyer**        **Mauricio Munoz**        **Prateek Katiyar**

**Anna Khoreva**        **Volker Fischer**

Bosch Center for Artificial Intelligence
lukas.hoyer@outlook.com        {firstname.lastname}@bosch.com

## S1    Further Details on Context Bias Detection on Synthetic Data

### S1.1    Synthetic Dataset

The proposed synthetic toy dataset consists of grayscale images of size $64 \times 64$ pixels, generated by combining digits from MNIST [1] with foreground and background textures from [2, 3], as can be seen in Fig. S1. Each MNIST image is bilinearly upsampled to $64 \times 64$ and used as a binary mask for the digit shape (with threshold $= 0.5$), which interior and exterior regions are filled with fore- and background textures, respectively. In total, a pool of $25$ textures are considered for fore- and background generation. To induce the texture variance among instances and make the segmentation task more challenging, we crop the textures randomly from the texture images of at least size $400 \times 400$ generated via [2]. One crop for each texture is visualized in Fig. S2. The texture variation between different random crops is shown in Fig. S3. For each synthetic image a corresponding segmentation ground truth is generated, where the MNIST digit defines the mask and the semantic class. Overall, $11$ semantic classes are considered, $10$ digits plus the background class.

For all dataset variants, a training and a test set are generated from the original MNIST training and validation sets respectively, using all $25$ textures. The training set contains 50k images and the test set consist of 1k images. For the training set, the seed for setting up the random generator can be varied in order to obtain different dataset variants. It specifies the digit order as well as the texture selection. Code to generate the toy dataset is provided under https://github.com/boschresearch/GridSaliency-ToyDatasetGen.

Figure S2: Random crop of each texture in the texture pool. The textures used as bias texture are shown in the first row (red framed).

Figure S1: Examples from the toy dataset.

Figure S3: Variance of textures across different random crops.

## S1.2 Experimental Setup

We utilize the U-Net [4] architecture with VGG16 [5], Mobilenetv2 [6], and ResNet18 [7] backbones, with a pixel-wise softmax layer (11 classes) as the final layer. For training, we use the multi-class cross entropy loss with the RMSPROP [8] optimizer. The networks are trained for 50 epochs, with the batch size 100; the learning rate is set to $10^{-3}$ and $10^{-4}$ for the first 40 and last 10 epochs, respectively.

To avoid adversarial artifacts in the estimation of the perturbation-based context saliency, we use the saliency maps of a lower resolution for optimization, i.e. $4 \times 4$ for $M_{context}$, and then upsample them to the original size of $64 \times 64$ using nearest neighbor upsampling for evaluation. We use a set of different constant perturbation grayscale values $\{0, 0.25, 0.5, 0.75, 1.0\}$ and for each image choose the one with the lowest loss value for $M = 0$, to avoid hurting the segmentation by having a low contrast between the object and the perturbation image. In order to avoid optimizing for the border artifacts of the network predictions, for the preservation loss computation, the request mask is eroded with a $3 \times 3$ kernel size.

The saliency maps are optimized using SGD with a momentum of $0.5$ and a learning rate of $0.2$ for 100 steps, starting with the initial map $M = 0.5$ and clipping saliency map values below $0.2$ to $0$ at each step in order to avoid artifacts. A weighting factor of $\lambda = 0.05$ is used (see Eq. 3). After the optimization, we check if an empty mask achieves a lower loss value than the optimized one and use the mask with the lowest loss as the final saliency map.

For the IG saliency method, the number of interpolation steps $n$ is set to 25 and for the SG saliency, we use 25 Gaussian noise samples with $\mu = 0, \sigma = 0.15(\max(x) - \min(x))$.

All computations were done on a Nvidia Titan Xp GPU with 12 GB memory.

## S1.3 Bias Detection

In this and the following section, we provide a more detailed analysis of the different saliency methods benchmarked on the toy dataset described in the main paper in Sec. 4. Along with our approach we additionally consider the Vanilla Gradient (VG) [9], Integrated Gradient (IG) [10], and SmoothGrad (SG) [11] saliency methods. Instead of aggregating the IoU, context bias detection (CBD) and context bias localization (CBL) metrics over multiple bias textures, we show them separately, to gain some insight how the bias texture may affect the performance of the context saliencies. The results are averaged over 5 different random seeds used for the training data generation.

We evaluate different networks trained on data biased towards a specific digit with a specific bias texture. When testing on the unbiased dataset, there is a drop in the segmentation IoU for the biased digit, which shows that the network has picked up the bias. As can be seen in Fig. S4 a) and S5 a) the extent of the drop is stable across different bias textures and is mostly affected by the bias digit.

For the different saliency methods, we have checked if this bias can be detected only using the biased dataset. For VG and SG (see Fig. S4 b/c) and S5 b/c)), the CBD highly deviates between the biased digits (diagonal), with the amount and direction heavily dependent on the bias texture. In practice, these methods are not applicable as there is no control over the bias texture. By design these gradient-based methods are more sensitive to high frequency patterns and thus lead to unfaithful explanations [12]. For IG and our context saliency (see Fig. S4 d/e) and S5 d/e)), we can observe a significantly smaller dependency on the bias texture allowing a bias detection to be independent from the bias texture. Moreover, for the weakly biased dataset, our grid saliency also roughly reflects the extent of the bias, which we observed in the segmentation IoU drop on the unbiased dataset (compare Fig. S5 a) and Fig. S5 e)).

Note that the mIoU drop for unbiased digits which look similar to the biased ones (e.g., four and nine) is not reflected in the generated saliency maps. By design, our method only looks for positive evidence (context that is present in the image and supports the classification) without taking into account negative evidence for the prediction. By applying an biased network to an unbiased dataset it is exposed to those negative biases as the bias texture acts as negative evidence for unbiased digits.

Figure S4: Context bias detection comparison of different saliency methods for strongly biased datasets. Instead of averaging over five different bias textures as done in Fig. 3, each bias texture is listed separately to show how the bias texture affects the different saliency methods. The bias detection performance is measured using the CBD metric (see Eq. 6).

Figure S5: Context bias detection comparison of different saliency methods for weakly biased datasets. Instead of averaging over five different bias textures as done in Fig. 3, each bias texture is listed separately to show how the bias texture affects the different saliency methods. The bias detection performance is measured using the CBD metric (see Eq. 6).

## S1.4 Bias Localization

A similar effect can be seen for bias localization in Fig. S6 and S7. While VG and SG (column a) and b)) highly depend on the bias texture causing the localization even to focus more on the unbiased half, IG and our method (column c) and d)) depend significantly less on the bias texture. However, IG only achieves a bias localization slightly above random guessing while our method is able to localize both strong and weak biases very well.

Figure S6: Bias localization across different grid saliency methods for strongly biased datasets. The CBL metric (see Eq. 7), averaged over five training dataset generation seeds, is shown with respect to bias texture and bias digit.

Figure S7: Bias localization across different grid saliency methods for weakly biased datasets. The CBL metric (see Eq. 7), averaged over five training dataset generation seeds, is shown with respect to bias texture and bias digit.

## S1.5    Network Comparison

In order to show that our grid saliency can be applied to several network architectures, we have repeated the experiments from Sec. 4.2 of the main paper with different backbones for Unet [4]. For that purpose, we have chosen VGG16 [5], ResNet18 (RN) [7], and MobileNetv2 (MN) [6] due to their different structure. All values are aggregated over five different bias textures and five training set generation seeds showing both the mean and the standard deviation. For the architecture comparison, only the weakly biased dataset variants are used.

**Segmentation mIoU on the unbiased test set.**    First, we have checked if the different architectures have picked up the bias from the dataset by applying network instances trained on biased datasets to the unbiased test set and calculated the segmentation mIoU. As can be see in Fig. S9 a), all architectures have a clear drop in mIoU for the biased digits (diagonal) in comparison to the baseline mIoU of the unbiased network (top row N). Therefore, we can conclude that all networks have picked up the bias. VGG achieves the best base segmentation performance and has also a smaller drop in mIoU for biased digits, meaning that it is less susceptible to a context bias. For that reason, we have chosen this configuration for the main paper.

**Bias Detection and Localization.**    Next, we have applied the context saliency methods to the different versions of the biased datasets to check if they are able to show the presence of a bias in the biased dataset itself.

For VG and SG (see Fig. S8), the mean over different bias textures is not susceptible to the bias, however, there is a high standard deviation for the biased digits (diagonal) caused by the dependence on the bias texture (see Sec. S1.3). IG and our method are both able to detect the context bias independent from the chosen network architecture. For VG, SG, and IG, we also observe different levels of focus on the context for different backbones, independent of the digit and if it is biased or not. For the bias localization, our method stably outperforms VG, SG, and IG across different backbones (see Fig. S10). For grid saliency we also observe a slight localization improvement for VGG over MN and RN, in contrast to other methods.

Figure S8: Comparison of the different grid saliency methods VG (top-left), SG (top-right), IG (bottom-left), and our perturbation saliency (bottom-right) across different network architectures (from top to bottom: VGG16 (VGG), Mobilenetv2 (MN), and ResNet18 (RN)). For each combination of grid saliency method and architecture, the mean and standard deviation (std) plots for context bias detection (CBD metric) are shown. All values are aggregated over five different bias textures and training set generation seeds.

Figure S9: Comparison of the segmentation mIoU on the unbiased test set for different network architectures: VGG16 (VGG), Mobilenetv2 (MN), and ResNet18 (RN). All values are aggregated over five different bias textures and training set generation seeds. Both the mean and the standard deviation (std) are visualized.

Figure S10: Comparison of localization capability for different network architectures ($x$ axis) using the CBL metric. All values are aggregated over ten bias digits, five different bias textures and five training set generation seeds. The error bars indicate the standard deviation over the bias digits.

## S1.6 Effect of Optimization Parameters

In Fig. S11 we report the effect of optimization parameters for grid saliency on context biased detection and localization performance (CBD, CBL). Influence of different optimization parameters on the quality of the obtained context explanations was evaluated on the basis of parameter sweeps around the chosen setting for the loss weighting $\lambda$, the learning rate and momentum, as well as initialization of the saliency mask (red points in Fig. S11). In our experiments in the main paper, all optimization parameters were set up by jointly looking at the two loss term values in Eq. 2, Sec. 3.1 and visual inspection of saliencies over a small image subset.

We notice that grid saliency shows comparable performance over a broad space of parameter settings (observing mostly smooth degradation with suboptimal parameter choices), with $\lambda$ clearly controlling the trade off between bias detection and localization quality (higher $\lambda$ value leads to a smaller salient region, see Sec. 3.1 of the main paper for method details). One sees, that the chosen parameter setting (red points in Fig. S11) balances well CBD and CBL performances (each high).

Figure S11: Effect of the grid saliency optimization parameters on the quality of context bias detection (mCBD) and localization (mCBL). mCBD (top row) and mCBL (bottom row) where computed for different settings of optimization parameters (from left to right): weighting lambda, learning rate, grey value initialization. All values are averaged over five randomly selected bias digits, five bias textures, and three training set generation seeds. The optimization parameters used in our experiments are depicted as red points. One sees, that the chosen parameter setting balances well mCBD and mCBL performances (each high). Note that for mCBD a high difference between biased and unbiased is beneficial.

## S2 Cityscapes Experiments

### S2.1 Experiment Setup

**Cityscapes Dataset.** A central motivation of this study is to better understand the behavior of segmentation networks. Cityscapes images describe rich scenes that make it arguably easy for segmentation models to learn clear context biases. We use the 500 (finely annotated) validation images available for Cityscapes as a starting point. We choose to analyze the validation data instead of training data in order to highlight behaviors that are likely to occur in a real world deployment scenario.

For the semantic segmentation, we have used the state-of-the-art network Deeplabv3+ [13] with a Mobilenetv2 backbone [6]. The weights were obtained from the original Deeplab repository[1].

**Implementation details.** The computation pipeline used to derive Fig. 6 in the main paper and its detailed version in Fig. S14 requires a selection of hyperparameters and design choices, which we explain here in more detail.

The saliency mask $M^*_{grid}$ is obtained by optimizing Eq. 2 with SGD, thus there is no guarantee of convergence to a global optimum. The loss function used in the optimization procedure (see in Eq.2) seeks to find a balance (partially controlled by $\lambda$) between penalizing the size of the produced saliency region and the preservation loss, which measures how well the softmax scores inside the request mask $R$ were restored to (at least) their initial values prior to perturbation, i.e. removing the image background. Typically, this preservation loss can be interpreted as a percentage relating to how much of the original softmax score was restored. So, for a loss of 0.1 or smaller, 90% or more of the original softmax activation scores must be restored. Samples that do not converge to a preservation loss of 0.1 or smaller are ignored in the computation. Similarly to the synthetic dataset, bilinear upsampling is used to upsample the optimized mask to the input image size.

The preservation loss in Eq. 2 is by definition normalized by the size of $R$, thus the size of $R$ does not directly influence the optimization convergence. In Fig. S12 we show the effect of the size of $R$, where context saliencies for each request mask $R$ were obtained with the same optimization parameters. Independent of the request mask $R$ size, for all riders salient context always falls on bikes.

Figure S12: Influence of the size of the request mask $R$ on context explanations provided by grid saliency. Independent of the request mask $R$ size, for all riders salient context always falls on bikes. Note that the context grid saliency explanations were calculated for each request region separately and are only combined together for the visualization.

a) Input frame

b) Softmax output on input frame

c) Perturbed context

d) Softmax output on peturbed context

e) Optimized image

f) Softmax output on optimized image

g) Optimized grid saliency map

h) Baseline fixed contour dilation

Figure S13: Intermediate visualizations for the rider class request mask on an input frame.

Fig. S13 shows some intermediate results for the analysis of a single frame, which illustrates the method and the optimization in a slightly more detailed way. The components include (a) the input frame, (b) the softmax scores for the "rider" class on the input frame, (c) the image with perturbed context given a request mask for the "rider" class, (d) the softmax output score of (c), (e) the optimized image (request mask plus learned background context), (f) the output softmax scores on (e) as an input, (g) the learned grid saliency, and (h) the dilation around the prediction used for the comparison to baseline.

Generally, penalizing the size of the saliency region is not sufficient to avoid spurious activations. We find that in general, the saliency mask at convergence represents a superset of the important pixels, i.e. it is often possible to slim down the saliency further without severely harming the preservation loss. Rather than adding regularizers directly in the loss function as [14], we simply clip all mask activation values smaller than or equal to 0.2 back to 0.0 for each step of the optimization. We find this leads to considerably less noisy activations and more focused and spatially coherent saliencies. Moreover, we manage potential border artifacts of the network predictions by eroding the request mask of each instance with a $3 \times 3$ erosion kernel.

Optimizing a low resolution saliency mask makes it hard to deal with small object instances in images. Specifically, a single pixel of the coarse mask already corresponds to a relatively wide spatial

context for small objects in the background of a scene. Because we count class labels in the activated saliency, this adds significant noise to the resulting statistics. In order to counteract this, we remove any connected components of the predicted object instances that have a pixel count smaller than $10k$ (or 625 for context explanations of erroneous predictions) from our request masks. This filters out a considerable number of images from the heatmap computation. Table S1 and Table S2 count the number of samples available for each class that pass the selection criteria used for the experiments.

The baseline metric dilates the contours of the predicted instances in a given image that pass the minimum size requirement using a total of $400k$ pixels for the dilated region. If multiple instances are present, these dilation pixels are shared amongst them. The number was chosen so as to be able to capture a meaningful portion of an object's spatial context, even if said object is large and in the foreground. An example baseline mask with the fixed dilation can be seen in Fig. S13 (h).

Table S1: Number of images in the Cityscapes validation dataset used to compute context grid saliency of the model predictions for each request class. For each request class, an image is included if and only if its computed grid saliency is not empty or invalid. This is the case only if the following conditions are met: 1. The request mask for the image contains at least one connected component larger than $10k$ pixels after applying the border erosion kernel. 2. The grid saliency computation does not converge to an empty context saliency mask. 3. The saliency optimization procedure converges with a preservation loss of 0.1 or less.

| Request (prediction) class | Number of samples |
|---|---|
| rider | 39 |
| car | 249 |
| person | 45 |
| bicycle | 105 |
| motorcycle | 16 |
| pole | 162 |
| traffic sign | 81 |

Table S2: Number of images in the Cityscapes validation dataset used to compute context grid saliency of the correct and erroneous model predictions. For each class, an image is included if and only if its computed grid saliency is not empty or invalid. This is the case only if the following conditions are met: 1. The request mask for the image contains at least one connected component larger than 625 pixels after applying the border erosion kernel. Here the request mask consists of the intersection between the ground truth specified in the first column and the model prediction specified in the second column. As these segments are typically much smaller and more disconnected, the above threshold was reduced to from $10k$ to 625 pixels. 2. The grid saliency computation does not converge to an empty context saliency mask. 3. The saliency optimization procedure converges with a preservation loss of 0.1 or less.

| GT class | Prediction class | Number of samples |
|---|---|---|
| rider | rider | 160 |
| | person | 67 |
| person | person | 217 |
| | rider | 45 |
| | car | 29 |
| bicycle | bicycle | 262 |
| | motorcycle | 12 |
| car | car | 323 |
| | motorcycle | 8 |
| | person | 33 |
| motorcycle | motorcycle | 44 |
| | bicycle | 18 |
| | car | 9 |
| pole | pole | 410 |
| | building | 324 |
| | vegetation | 145 |

## S2.2 Further Quantitative Examples

Fig. S15 shows additional representative qualitative examples of our method for a variety of request classes. Some interesting effects include saliency activations on road signs marked on the pavement for the car class (Fig. S15 (e)), as well as the behavior of the method for occluded objects. As can be seen in subfigures (b) and (d), the grid saliency for occluded objects tends to activate more strongly along the entire contour of the object. In particular, subfigure (d) contains two objects of similar size, pose and background, where only the second is partially occluded.

In certain cases, the prediction of the image with completely perturbed context is already so good so that no gradients are available for the preservation loss, and the optimization then focuses on minimizing the total loss by simply getting rid of the saliency activation altogether. This typically results in large object instances of clearly identifiable classes such as cars or pedestrians with no saliency activations. Fig. S15 i) illustrates this for the left and center group of pedestrians. We consider this effect in itself to be a valid solution. An empty context saliency means that the requested object is self-containing and context is not necessary for its segmentation.

## Grid Saliency

| Request Class \ Context Class | road | building | vegetation | sidewalk | car | bicycle | motorcycle | person | rider | pole | traffic_sign | traffic_light | bus | train | truck | sky | terrain | fence | wall |
|---|---|---|---|---|---|---|---|---|---|---|---|---|---|---|---|---|---|---|---|
| car | 0.409 | 0.103 | 0.082 | 0.085 | 0.066 | 0.011 | 0.011 | 0.018 | 0.005 | 0.023 | 0.007 | 0.001 | 0.010 | 0.000 | 0.017 | 0.002 | 0.022 | 0.011 | 0.005 |
| bicycle | 0.156 | 0.158 | 0.045 | 0.248 | 0.024 | 0.096 | 0.007 | 0.035 | 0.061 | 0.035 | 0.002 | 0.000 | 0.000 | 0.001 | 0.000 | 0.000 | 0.014 | 0.014 | 0.010 |
| motorcycle | 0.216 | 0.035 | 0.017 | 0.224 | 0.027 | 0.040 | 0.169 | 0.032 | 0.067 | 0.050 | 0.002 | 0.000 | 0.002 | 0.000 | 0.000 | 0.000 | 0.000 | 0.010 | 0.000 |
| person | 0.172 | 0.284 | 0.066 | 0.165 | 0.024 | 0.044 | 0.002 | 0.073 | 0.006 | 0.034 | 0.005 | 0.003 | 0.000 | 0.000 | 0.000 | 0.000 | 0.015 | 0.011 | 0.003 |
| rider | 0.137 | 0.093 | 0.093 | 0.096 | 0.066 | 0.282 | 0.061 | 0.007 | 0.073 | 0.012 | 0.008 | 0.002 | 0.003 | 0.001 | 0.000 | 0.001 | 0.005 | 0.001 | 0.009 |
| pole | 0.052 | 0.145 | 0.254 | 0.197 | 0.021 | 0.010 | 0.007 | 0.028 | 0.002 | 0.066 | 0.013 | 0.005 | 0.000 | 0.002 | 0.000 | 0.008 | 0.037 | 0.019 | 0.007 |
| traffic_sign | 0.038 | 0.273 | 0.208 | 0.024 | 0.086 | 0.028 | 0.011 | 0.016 | 0.001 | 0.066 | 0.095 | 0.003 | 0.002 | 0.000 | 0.000 | 0.038 | 0.009 | 0.021 | 0.010 |

## Baseline

| Request Class \ Context Class | road | building | vegetation | sidewalk | car | bicycle | motorcycle | person | rider | pole | traffic_sign | traffic_light | bus | train | truck | sky | terrain | fence | wall |
|---|---|---|---|---|---|---|---|---|---|---|---|---|---|---|---|---|---|---|---|
| car | 0.373 | 0.217 | 0.179 | 0.042 | 0.016 | 0.009 | 0.002 | 0.014 | 0.004 | 0.021 | 0.009 | 0.003 | 0.005 | 0.001 | 0.007 | 0.016 | 0.010 | 0.006 | 0.004 |
| bicycle | 0.314 | 0.239 | 0.101 | 0.128 | 0.032 | 0.018 | 0.004 | 0.028 | 0.018 | 0.021 | 0.008 | 0.002 | 0.004 | 0.001 | 0.000 | 0.004 | 0.005 | 0.005 | 0.010 |
| motorcycle | 0.347 | 0.179 | 0.144 | 0.093 | 0.044 | 0.021 | 0.022 | 0.016 | 0.014 | 0.030 | 0.015 | 0.001 | 0.004 | 0.000 | 0.000 | 0.004 | 0.000 | 0.009 | 0.000 |
| person | 0.276 | 0.293 | 0.122 | 0.099 | 0.043 | 0.012 | 0.002 | 0.019 | 0.003 | 0.024 | 0.008 | 0.003 | 0.001 | 0.001 | 0.000 | 0.007 | 0.006 | 0.007 | 0.004 |
| rider | 0.289 | 0.195 | 0.202 | 0.078 | 0.038 | 0.056 | 0.006 | 0.014 | 0.013 | 0.018 | 0.009 | 0.003 | 0.010 | 0.000 | 0.001 | 0.013 | 0.006 | 0.002 | 0.004 |
| pole | 0.186 | 0.215 | 0.201 | 0.128 | 0.024 | 0.009 | 0.002 | 0.016 | 0.003 | 0.024 | 0.014 | 0.006 | 0.002 | 0.002 | 0.000 | 0.024 | 0.022 | 0.014 | 0.011 |
| traffic_sign | 0.110 | 0.315 | 0.215 | 0.051 | 0.072 | 0.014 | 0.002 | 0.020 | 0.003 | 0.032 | 0.014 | 0.004 | 0.001 | 0.000 | 0.000 | 0.032 | 0.009 | 0.022 | 0.009 |

## Difference Grid Saliency and Baseline

| Request Class \ Context Class | road | building | vegetation | sidewalk | car | bicycle | motorcycle | person | rider | pole | traffic_sign | traffic_light | bus | train | truck | sky | terrain | fence | wall |
|---|---|---|---|---|---|---|---|---|---|---|---|---|---|---|---|---|---|---|---|
| car | 0.036 | -0.113 | -0.097 | 0.043 | 0.050 | 0.003 | 0.009 | 0.004 | 0.001 | 0.002 | -0.002 | -0.002 | 0.006 | -0.001 | 0.010 | -0.014 | 0.013 | 0.005 | 0.001 |
| bicycle | -0.158 | -0.081 | -0.056 | 0.120 | -0.008 | 0.078 | 0.003 | 0.006 | 0.042 | 0.015 | -0.005 | -0.002 | -0.004 | -0.000 | -0.000 | -0.004 | 0.009 | 0.009 | 0.000 |
| motorcycle | -0.130 | -0.144 | -0.127 | 0.131 | -0.017 | 0.019 | 0.147 | 0.015 | 0.053 | 0.020 | -0.013 | -0.001 | -0.002 | 0.000 | -0.000 | -0.004 | -0.000 | 0.001 | 0.000 |
| person | -0.104 | -0.009 | -0.055 | 0.066 | -0.019 | 0.032 | 0.000 | 0.054 | 0.004 | 0.010 | -0.003 | 0.000 | -0.001 | -0.001 | -0.000 | -0.007 | 0.009 | 0.004 | -0.001 |
| rider | -0.152 | -0.103 | -0.109 | 0.017 | 0.027 | 0.226 | 0.054 | -0.006 | 0.060 | -0.006 | -0.001 | -0.000 | -0.008 | 0.001 | -0.001 | -0.012 | -0.001 | -0.000 | 0.005 |
| pole | -0.134 | -0.069 | 0.053 | 0.068 | -0.003 | 0.001 | 0.005 | 0.012 | -0.001 | 0.042 | -0.000 | -0.001 | -0.002 | 0.000 | -0.000 | -0.015 | 0.015 | 0.005 | -0.004 |
| traffic_sign | -0.071 | -0.041 | -0.007 | -0.027 | 0.013 | 0.014 | 0.009 | -0.003 | -0.002 | 0.034 | 0.081 | -0.000 | 0.000 | -0.000 | -0.000 | 0.006 | 0.001 | -0.001 | 0.001 |

Figure S14: Comparison of saliency distributions for a selected set of classes across the Cityscapes validation data. Note that the rows of the *Grid Saliency* and *Baseline* maps do not sum up to 1, because only a subset of classes is considered. S14

a) Motorbike     b) Bicycle     c) Bicycle Group

d) Bicycle     e) Car     f) Car Group

g) Car     h) Large Car     i) Pedestrian Groups

j) Pedestrian Group     k) Pedestrian Group     l) Pedestrian

m) Poles     n) Pole     o) Pole

p) Pole     q) Cycler     r) Cyclers

s) Cyclers     t) Cycler     u) Traffic Sign

v) Traffic Sign     w) Traffic Sign     x) Traffic Sign

Figure S15: Cityscapes examples supporting the class statistics.

## S3   Additional Results on MS COCO

In order to show that our grid saliency can be applied on different types of real data and to various network architectures, we have repeated the experiments from Sec. 4.2 of the main paper on the MS COCO images [15] with different backbones for the state-of-the-art Deeplabv3+ [13] network. In particular, we have chosen MobileNetv2 [6] and Xception (XC) [16] as backbones due to their different structure. All implementation details and optimization parameter settings are borrowed from the Cityscapes experiments.

In Fig. S16, we show some examples of context explanations on MS COCO obtained with our grid saliency method. We observe that grid saliency provides sensible and coherent explanations for network decision making, which reflect semantic dependencies present in the data (e.g. boat appears on the water). Context explanations produced by grid saliency can be also utilized to compare architectures with respect to their capacity to either learn or to be invariant towards context. E.g., in Fig. S16 the segmentation network with MobileNetv2 (MN) backbone learnt to rely more on context in contrast to its variant with a more powerful Xception (XC) backbone, which, for example, does not look at rails as context to segment the train.

Figure S16: Context explanations produced by grid saliency on MS COCO [15] for the Deeplabv3+ [13] network with MobileNetv2 [6] and Xception (XC) [16] as backbones.

## Footnotes

[1]http://download.tensorflow.org/models/deeplabv3_mnv2_cityscapes_train_2018_02_05.tar.gz