[Reviews · NeurIPS 2019]

Reviewer 1



1. Grid saliency can produce spatially coherent explanations for dense prediction networks firstly. This work seems to be sufficiently innovative and meaningful. 2. Experiment results are clear in the context biases research on synthesis dataset. However, a comparative analysis should be more detailed. The grid saliency is based on perturbation saliency method, while other contrast methods are belong to gradient-based saliency methods. I have doubts about the relationship between these methods. ------------------------------------ The author responses provided a comparative analysis of two methods. 3. For the real-world dataset cityscapes, more quantitative results are expected to be provided in the submission like the numerical results in supplementary material. I am also interested in how to use context explanations of erroneous predictions to improve the performance. --------------------------- The author responses provided quantitative experiment results and more details about the practice.

Reviewer 2



1. The major concern with this works is that the impact of the work seems to be limited, and I am not sure if the practitioners would be interested in grid saliency and how would this be useful for them. 2. The explanation of how to obtain grid saliency map is not clear. For example, in equation 2, the M*_grid is not clear how to obtain from the optimization. How would the size of R affect the M? Would we always be able to obtain an optimal M*_grid given any request mask R? It is also not very clear how to twist the lambda to make this works.

Reviewer 3



ORIGINALITY To the best of my knowledge, there has been no previous study on deep neural networks about explaining pixel-wise predictions in image segmentation. QUALITY The reported results indicate the superiority of this perturbation-based method with respect to three gradient-based methods. Results seem convincing. CLARITY The text is clearly written, with the necessary visualizations and results, all of them commented to guide the reader's understanding. SIGNIFICANCE TRhe work makes an interesting and necessary step that will help the community working on this task, and any other pixel-wise one.

[Author Response · NeurIPS 2019]

We thank the reviewers for their valuable feedback on our work, indicating its novelty (R1,R3) and effectiveness
(R1,R3), acknowledging the potential interest and utilization of Grid Saliency in the explainability community (R2,R3).

**R1** and **R2** raised a point about the practical impact of our work: how practitioners would use Grid Saliency (GS) and,
in particular, how to use its context explanations of error cases to improve performance. We motivate its utility for dense
prediction networks (which is novel) for the following applications: *1) Architecture comparison:* Context explanations
produced by GS can be used to compare architectures wrt. their capacity to either learn or to be invariant towards
context. E.g., in Fig. 1a the segm. network with MobileNet (MN) backbone learnt to rely more on context in contrast to
its variant with a more powerful Xception (XC) backbone, which can correctly predict train w/o looking at rails. *2)*
*Network generalization via active learning:* Existing context biases might impair network generalization. E.g., cows
might mostly appear on grass during training. A network that was trained and evaluated on this data and picked up that
bias will perform poorly in real-world cases, where the cow, for example, appears on road (in Fig. 1b top right, the cow
gets misclassified as horse). Here, removing all context yields a correct classification (Fig. 1b bottom row) and analysis
of the context explanations (Fig. 1b top left) produced by GS shows responsible context for the erroneous classification.
Now, actions can be taken, such as targeted extra data collection. *3) Adversarial detection:* GS can be used to detect
and localize adversarial patches outside object boundaries (e.g. Lee and Kolter [2019]). Cases for which the salient
regions lie largely outside an object, would strongly indicate the presence of an adversary or misguided prediction.

**R1** [Relationship between gradient and perturbation methods]: We agree that gradient- and perturbation-based saliency
methods use different techniques. However, both aim to compute a relevance map for an input. We compared these
maps for different methods to evaluate how well they could detect and localize relevant input parts (controlled by our
synthetic data). Hence, wrt. the property of indicating relevant input parts, we think the two techniques are comparable
and next discuss their more detailed comparison.

**R1** [More comparative analysis on synthetic
dataset]: For more detailed analysis, we refer to
Sec. S1.3-S1.5 in sup. mat. From Fig. S4 and
S6 we observed that for context bias detection
and localization, respectively, gradient methods
are prone to high variations dependent on the
background texture choice, as by design these

(a) Context explanations on COCO.  (b) Error case on COCO.  (c) Effect of $R$ size.

Figure 1: Qualitative examples.

methods are more sensitive to high frequency patterns and thus lead to unfaithful explanations Adebayo et al. [2018]. In
contrast, perturbation-based GS can consistently detect and localize context bias independent of texture choice (partially
due to perturbing larger image regions). We also compare gradient and perturbation methods across different networks
in Fig.S9-10, confirming the superior performance of the perturbation GS. We will add these findings to Sec.4.2.

**R1** [Quant. results on Cityscapes]: We complemented the quantitative
results in suppl. Sec. S2.2 (Fig. S11) with an analysis of context expla-
nations on erroneous predictions. Tab. 1 shows the case where rider was
misclassified as person. We used the intersection of GT mask and (error)

Table 1: Context class statistics of errors.

| | | Context class | | | | | | |
|---|---|---|---|---|---|---|---|---|
| GT | Pred.($R$) | road | bike | veg. | build. | sidew. | rider | person |
| rider | person | 0.20 | 0.06 | 0.22 | 0.21 | 0.06 | 0.05 | 0.01 |
| rider | rider | 0.15 | 0.30 | 0.08 | 0.09 | 0.10 | 0.07 | 0.01 |
| person | person | 0.11 | 0.09 | 0.09 | 0.28 | 0.16 | 0.00 | 0.07 |

prediction as request mask $R$, and similarly to Fig. 5 and Fig. S11 computed semantic class statistics inside the salient
context. Note that for correctly classified riders context saliency mostly focuses on bike (30%), but is almost non-present
(6%) when rider is mistaken as person. A detailed quantitative analysis with more error cases will be added to the paper.

**R2** [$M^*_{grid}$ computation, effect of $R$ and optimiz. parameters]: $M^*_{grid}$ is
obtained by optimizing Eq.2 with SGD (see Sec. S1.2, S2.1 in sup. mat.),
thus there is no guarantee for global convergence. The loss function in
Eq.2 aims to find a balance (partially controlled by $\lambda$) between penalizing
the salient region size and the preservation loss, which measures how well
the softmax scores inside the request mask $R$ were restored to their initial

Figure 2: Effect of optimization parameters
on the synthetic dataset. Red points depict the
parameters used in our experiments.

values, prior to perturbation. This loss is by definition normalized by the $R$ size, thus the size of $R$ doesn't directly
influence the optimization convergence. In Fig. 1c we show the effect of $R$ size, where saliencies for each $R$ were
obtained with the same optimiz. parameters. Independent of $R$ size, for all riders salient context always falls on bikes.
In Fig.2 we report the effect of optimization parameters (learning rate, $\lambda$, mask initialization) on context biased detection
and localization performance (CBD, CBL). GS shows comparable performance over a broad space of parameter
settings (experiencing smooth degradation with suboptimal parameter choices), with $\lambda$ clearly controlling the trade off
between bias detection and localization quality (higher $\lambda$ value leads to a smaller salient region, see L130-136). In our
experiments the optimization parameters (red points in Fig.2) were set up by jointly looking at the two loss term values
in Eq.2 and visual inspection of saliencies over a small image subset. We will add this discussion to the sup. mat.

**R3** [Results on other dataset]: We agree with R3 on evaluating GS on different segmentation datasets. In Fig. 1a, we
show some first examples of context explanations on COCO, which we will add to and discuss in the paper.

**R3** [Literature]: We will add the literature on the importance of context Uijlings et al. [2012], Azaza et al. [2018].

[Meta-Review · NeurIPS 2019]

This is a good paper and I recommend to accept it. Although technical novelty of this submission is not amazing, I think the problem analyzed is interesting and of practical importance. The research is also well executed and the paper is written well.